# The Role of Pharmacogenetic-Based Pharmacokinetic Analysis in Precise Breast Cancer Treatment

**DOI:** 10.3390/pharmaceutics16111407

**Published:** 2024-10-31

**Authors:** Xinyu Wu, Huihua Xiong

**Affiliations:** Department of Oncology, Tongji Hospital, Tongji Medical College, Huazhong University of Science and Technology, Wuhan 430074, China; wuxy@hust.edu.cn

**Keywords:** pharmacokinetics, pharmacogenomics, breast cancer, precise treatment

## Abstract

Given the high prevalence of breast cancer and the diverse genetic backgrounds of patients, a growing body of research emphasizes the importance of pharmacogenetic-based pharmacokinetic analysis in optimizing treatment outcomes. The treatment of breast cancer involves multiple drugs whose metabolism and efficacy are influenced by individual genetic variations. Genetic polymorphisms in drug-metabolizing enzymes and transport proteins are crucial in the regulation of pharmacokinetics. Our review aims to investigate the opportunities and challenges of pharmacogenomic-based pharmacokinetic analysis as a precision medicine tool in breast cancer management.

## 1. Introduction

Globally, over 2.30 million new cases of breast cancer are diagnosed and cause 665,684 deaths annually [1]. BC is a highly heterogeneous disease at the molecular level, predominantly sporadic, affecting people without a known familial cancer history. Key risk factors for BC include advanced age, obesity, smoking, radiation exposure, excessive alcohol consumption, and hormone replacement therapy [2,3]. Additionally, mutations in certain genes, whether inherited or acquired, such as BRCA1 and BRCA2 (breast cancer susceptibility genes 1 and 2), can lead to the development of BC [4]. The genetic landscape of BC is remarkably diverse, with all patients presenting unique genetic characteristics [5].

Precision medicine has become the cornerstone of breast cancer treatment [6]. Routine diagnostics now integrate with molecular markers, dividing patients into Luminal, HER2-overexpressed, and basal-like [7], which guides treatment decisions on endocrine therapy, chemotherapy, or molecularly targeted therapy [8]. By considering genetic, environmental, and lifestyle elements, precision medicine seeks to enhance the treatment effectiveness and reduce side effects [9,10].

Within this broad framework, the critical role of pharmacogenetic (PGx) variations is frequently overlooked, especially when talking about drug metabolism [11]. Genetic factors contribute significantly to pharmacokinetic variability, particularly affecting the enzymes and transporters that regulate drug absorption, distribution, metabolism, and excretion. Understanding how an individual’s genetic profile influences drug metabolism, further for its efficacy and toxicity, facilitates personalized drug selection and dosing strategies to improve effectiveness and minimize adverse reactions.

The FDA defines pharmacogenomic biomarkers as tools for personalizing drug treatments [12,13]. These biomarkers include genetic variants associated with specific drug responses and guide treatment management by providing information on the potential impact of drugs on safety, effectiveness, and pharmacokinetics [14]. They are selected based on clinical studies from various regions to ensure their reliability and efficacy [15].

Biomarkers, including genetic variants, are critical for assessing individual responses to particular medications. They provide valuable information for healthcare professionals to assess the potential impact of drugs on safety, effectiveness, and pharmacokinetics. These biomarkers are chosen based on clinical studies conducted across different regions to ensure their reliability and efficacy.

This article aims to review the current roles of pharmacokinetic analysis in treating breast cancer and investigate potential future developments within this area. By analyzing genetic variations, we can move towards more personalized and effective treatment strategies, potentially improving outcomes for breast cancer patients.

## 2. Metabolic Enzymes and Genes Related to Breast Cancer Drug Metabolism

Individual variations in drug efficacy and adverse reactions are influenced by multiple factors, with genetic variations contributing approximately 20–25% to these differences, though this varies by drug [16]. Genetic influences include both common and rare single nucleotide polymorphisms (SNPs) and structural variations (SVs), although the mechanisms underlying many of these genetic variations remain unclear [17,18]. SNPs arise from variations in individual DNA bases and can differ between individuals and across different races and ethnicities. 

SNPs are classified based on their phenotypic impact [19]. Most of the population is considered a normal metabolizer (NM) with typical enzyme activity. Poor metabolizers (PM) show minimal or no enzyme activity, while ultra-rapid metabolizers (UM) exhibit heightened enzyme activity. Intermediate metabolizers (IM) display enzyme activity that falls between those of PM and NM [20]. The phenotypic effects of SNPs on drug-metabolizing enzymes can range wisely from mild to significantly altering enzyme activity [20,21]. In the previously published literature, there has been inconsistent reporting of phenotype variability across ethnicities and laboratories [21].

In breast cancer, gene products are essential in the action of anticancer drugs. The cytochrome P450 (CYP450) enzyme family is crucial in the oxidation or conjugation of xenobiotics. This process assists in transforming drugs into forms that can be excreted more easily through the kidneys [22]. Additionally, the multidrug resistance proteins (MDRPs), part of the ATP-binding cassette (ABC) transporters superfamily, are responsible for anti-drug efflux and influencing both the pharmacokinetics and pharmacodynamics of medications [23,24]. Notable examples include ABCB1 (P-glycoprotein, P-gp) and ABCG2 (breast cancer resistance protein, BCRP) [25]. SNPs in CYP450 enzymes and MDRPs can alter the pharmacokinetic (PK) and pharmacodynamic (PD) profiles of drugs used in cancer treatment [22,23]. Germline PGx variations affect the PK/PD characteristics of antitumor drugs, potentially leading to increased toxicity or reduced efficacy in specific patient subgroups [26]. Although preventive germline PGx testing remains atypical in oncology, its potential is significant, particularly with the expanding utilization of next-generation sequencing (NGS) and the growing number of germline mutation discoveries [26].

## 3. Pharmacogenetic Variants and Breast Cancer Drugs

This section explores the pharmacokinetics of drugs used in breast cancer treatment, focusing on three primary therapeutic approaches: endocrine therapy, chemotherapy, and anti-HER2 targeted therapy. Genetic variations can affect the effectiveness of treatments and their associated toxicity by modifying the pharmacokinetics of drugs, which is crucial for dose adjustments and preventive strategies, especially for non-targeted drugs. Key variations involved in the metabolism, transport, and excretion of drugs are listed in Figure 1. A detailed summary of these pharmacogenetic variants and their impact on drug metabolism and efficacy is provided in Table 1.

### 3.1. Endocrine Therapy

#### 3.1.1. Tamoxifen

The relationship between CYP2D6 genotype and tamoxifen efficacy is among the most extensively studied pharmacogenetic issues in breast cancer treatment [27]. Tamoxifen, a selective estrogen receptor modulator (SERM), is widely used for treating and preventing estrogen receptor (ER)-positive breast cancer [28]. It functions by antagonizing estrogen signaling in ER-dependent cancer cells [29]. As a pro-drug, tamoxifen needs to be activated by CYP2D6 to form endoxifen, its most active metabolite, which has a potency 100 times greater than the parent compound [29].

The CYP2D6 gene is widely acknowledged as a key pharmacogenetic predictor in drug pharmacokinetics [30,31]. It encodes an enzyme that exhibits high polymorphism, with more than a hundred variants identified, including frequently observed gene duplications and deletions. This variability leads to significant differences in CYP2D6 activity among individuals. According to the Clinical Pharmacogenetics Implementation Consortium (CPIC) guidelines, over 20 of these variants are categorized as loss-of-function, 12 as having reduced function, and about 10 as having standard function [32].

Early studies in 2005 suggested optimizing tamoxifen therapy based on CYP2D6 genotyping [33]. Still, debates on its clinical significance persisted until the CPIC recommended CYP2D6 genotyping [34]. While ESMO stated in 2019 expressed skepticism in 2019 about its clinical utility, highlighting the challenges of translating pharmacogenetic findings into practice. The KARISMA trial [35], reported in 2021, indicates that ultra-rapid CYP2D6 metabolizers (CYP2D6-UM) experienced more severe tamoxifen-associated vasomotor symptoms, such as hot and cold flashes; and neuropsychiatric symptoms, including mood swings and irritability. These symptoms are strongly linked to reduced quality of life. As a result, one-third of CYP2D6-UM discontinued tamoxifen therapy within one month, a rate significantly higher than that observed in poor metabolizers [36]. Furthermore, selective serotonin reuptake inhibitors (SSRIs), which are commonly prescribed to alleviate tamoxifen-induced hot flashes, are potent CYP2D6 inhibitors [37]. Studies have demonstrated that concomitant use of SSRIs, particularly paroxetine, with tamoxifen is associated with poorer clinical outcomes [37,38,39]. It can be assumed that CYP2D6-UM patients who already metabolize tamoxifen at a faster rate may experience additional inhibition from SSRIs, further impairing tamoxifen activation. 

These findings may explain the higher breast cancer-specific mortality observed in patients with CYP2D6-UM compared to less efficient metabolizers, suggesting that genetic background significantly impacts adherence to tamoxifen therapy.

Additionally, CYP2D6-UM shows a more pronounced reduction in mammographic density than poor metabolizers, which may suggest a potentially greater therapeutic effect of tamoxifen in this group [35]. When analyzed by menopausal status, a significant association between CYP2D6 metabolizer status and mammographic density reduction was observed in premenopausal women but not in the postmenopausal group [35]. This discrepancy aligns with previous studies [40] suggesting that the impact of CYP2D6 on tamoxifen outcomes might differ based on menopausal status. The reasons for this variation remain unclear.

A prospective study found that carriers of the CYP2D6*10 heterozygous genotype experienced a notable reduction in 5-year disease-free survival (DFS) [41]. This CYP2D6*10 variant may thus serve as a significant prognostic indicator for DFS. The U.S. Food and Drug Administration (FDA) recommends CYP2D6 genotype testing for estrogen receptor-positive breast cancer patients before initiating tamoxifen treatment to optimize drug efficacy [13].

#### 3.1.2. Aromatase Inhibitors (AIs)

Aromatase, encoded by the CYP-19 gene within the cytochrome P450 superfamily, catalyzes the conversion of peripheral androgens into estrogens. AIs lower plasma estrogen levels by suppressing aromatase activity, thereby inhibiting the hormone-dependent growth of breast tumor cells.

Patient responses to AIs may vary due to genetic differences in the CYP19 family, which only consists of CYP19A1 [42]. Studies have shown that SNPs in the CYP19A1, such as rs6493497 and rs7176005, are linked to higher baseline aromatase activity in postmenopausal breast cancer patients undergoing neoadjuvant AI therapy [43]. However, carriers of these mutations often present higher E2 plasma levels before and after treatment in the adjuvant setting [43]. Additionally, Ghimenti et al. found no significant correlation between these mutations and the efficacy of AI treatment [44]. Assuming that AI therapy might not be able to reduce estrogen levels in patients carrying these SNPs by suppressing the increased aromatase activities. Conversely, Liu et al. observed that patients with the CYP19A1 rs4646 SNP mutation exhibited significantly longer progression-free survival (PFS) and overall survival (OS) compared to those with wild-type alleles, suggesting this mutation as a potential prognostic marker for anastrozole response in breast cancer patients. 

To fully understand the impact of genetic differences in metabolic enzymes on drug response, it is essential to delineate the primary metabolic pathways of the drugs in question. Anastrozole and letrozole are non-steroidal AIs that inhibit the aromatase pathway by reversibly binding to its active site. 

Anastrozole is primarily metabolized by CYP3A4, with CYP3A5 and CYP2C8 playing secondary roles, converting it to hydroxy anastrozole, which is subsequently transformed into anastrozole N-glucuronide by UGT1A4 [45,46]. Letrozole undergoes primary metabolism by CYP3A4 and CYP2A6 to methanol [46], followed by conjugation with UGT2B7 [47]. Letrozole has a high affinity for CYP2A6 but also has a great chance to inhibit it [48]; at that time, the metabolic pathway shifts to the CYP3A4 route [49]. Polymorphisms in CYP2A6 and CYP3A subtypes can significantly alter enzyme activity, influencing drug metabolism. For instance, intermediate and slow CYP2A6 metabolizers have been found to have elevated letrozole plasma concentrations [50]. Notably, polymorphisms in CYP2A6 show variation across ethnicities, with a greater percentage of slow metabolizers in Asian communities (10–20%) in contrast to Caucasians (1.2%) [51].

The activity of glucuronosyltransferases, particularly UGT1A4 and UGT2B7, may also be modulated by genetic polymorphisms, leading to substantial interindividual variability in drug metabolism [47,52]. For instance, UGT1A4-mediated anastrozole glucuronidation is associated with therapeutic outcomes in breast cancer patients [53]. The presence of SNPs in the coding and non-coding regions of UGT1A can influence the binding activity of the UGT1A family [53,54]. These SNPs may not only affect treatment response but also relate to drug toxicity and adverse effects [55].

Exemestane, a steroidal AI, irreversibly inhibits aromatase activity by covalently binding to aromatase, thereby reducing estrogen synthesis. Exemestane is primarily metabolized to 17-hydroxy exemestane (MI) via CYP1A1/2 and CYP4A11 or to 6-hydroxymethyl exemestane (MII) mediated by CYP3A [56]. The main metabolic pathway for both exemestane and MI involves glutathionation via glutathione-S-transferase (GST) [57].

A prospective study showed that CYP3A4*22 was significantly associated with higher concentrations of exemestane (*p* < 0.01), with elevated levels observed in Caucasians, patients with elevated transaminases, those with renal impairment, individuals with a lower body mass index, and patients who had not undergone chemotherapy (all *p* < 0.05) [58]. The GSTA1 *B*B genotype was associated with reduced conjugated metabolites of exemestane in plasma, suggesting inhibited metabolism [59]. However, the association between exemestane efficacy and drug-metabolizing enzyme genotypes remains inconclusive, necessitating further research.

#### 3.1.3. Cyclin-Dependent Kinase (CDK) 4/6 Inhibitor

The CDK 4/6 complex regulates the cell cycle’s transition from the G1 to the S phase [60]. Abnormal activation or overexpression of this complex can lead to uncontrolled cell proliferation and contribute to cancer development. CDK4/6 inhibitors effectively suppress tumor cell proliferation by blocking this critical regulatory point. When used alongside endocrine therapy, CDK4/6 inhibitors have become an important treatment modality for breast cancer, with approved drugs including palbociclib, ribociclib, abemaciclib, and dalpiciclib [61,62,63,64].

These inhibitors share similar pharmacokinetic profiles, rapidly absorbed and distributed, with metabolism primarily occurring in the liver [65,66,67,68]. The CYP3A4 enzyme plays a central role in their metabolism. Clinical studies have shown that the concurrent use of strong CYP3A4 inhibitors (e.g., itraconazole and ritonavir) markedly elevates the plasma concentrations of CDK4/6 inhibitors, thereby raising the risk of toxicity. Consequently, the FDA advises against combining CDK4/6 inhibitors with strong CYP3A4 inhibitors [69,70,71].

Although research on the impact of genetic polymorphisms in the CYP3A family on CDK4/6 inhibitor metabolism is limited, it is plausible that variations in CYP3A4 and CYP3A5 could lead to differences in drug metabolism, potentially influencing the pharmacokinetics and clinical outcomes of these drugs. Future studies should focus on exploring the effects of these genetic variations to optimize individualized treatment strategies.

### 3.2. Chemotherapy

#### 3.2.1. Taxanes

Taxanes, such as paclitaxel and docetaxel, are cornerstone treatments for breast cancer. The therapeutic impact of these agents is realized by attaching to the β-subunit of microtubule proteins, disturbing the function of microtubules, preventing mitosis, and ultimately inducing cell demise. Paclitaxel undergoes primary metabolism in the liver, where it is acted upon by the enzymes CYP2C8 and CYP3A4, resulting in the formation of hydroxylated metabolites. Transport proteins, including ABCB1 (P-glycoprotein), ABCC2, and SLCO1B1, mediate the transmembrane transport of drugs, further influencing their tissue distribution and excretion [72,73,74].

Genetic variations in CYP2C8 and the CYP3A family can impact the metabolism and efficacy of paclitaxel. As early as 2002, researchers reported that non-synonymous CYP2C8 variants could potentially reduce paclitaxel metabolic activity [75]. A prospective cohort study from 2012 found that individuals with the CYP2C8*3 variant had a higher chance of experiencing complete remission following neoadjuvant paclitaxel treatment [76]. According to a Danish cohort study involving premenopausal women diagnosed with non-metastatic breast cancer, carriers of the GSTP1 rs1138272 and CYP3A rs10273424 genetic variants exhibited elevated mortality rates. While individuals with the SLCO1B1 rs2306283 variant demonstrated a reduced risk of cancer recurrence and mortality [74]. In another study focusing on locally advanced breast cancer patients, those with low circulating CYP3A4 mRNA levels showed higher response rates to docetaxel compared to patients with elevated CYP3A4 mRNA levels (*p* < 0.01) [77].

The toxicity associated with paclitaxel remains a massive challenge in breast cancer treatment. The presence of genetic variations in drug-metabolizing enzymes including CYP2C8 and CYP3As, along with transporters like ABCB1, contributes to considerable interindividual variability in pharmacokinetic patterns and reactions to toxicity among individuals [78,79,80,81,82,83,84,85]. Notably, patients with the ABCB1-2677GT and CYP2C8*1*3 genotypes exhibit markedly reduced paclitaxel clearance compared to those with ABCB1-2677GT and CYP2C8*1*1 genotypes (*p* = 0.032) [86].

The CYP2C8 *3 variant has been pinpointed in several studies as a potential contributor to the development of chemotherapy-induced peripheral neuropathy (CIPN) associated with paclitaxel [76,87]. It is reported that individuals carrying low-metabolism variants (CYP2C8*2, CYP2C8*3, or CYP2C8*4) experienced grade 2 or higher neuropathy at lower cumulative paclitaxel doses [88]. Further, Demurtas et al. confirmed that CYP3A4 and CYP2C8 SNPs may modulate paclitaxel toxicity [89]. Gudur et al. discovered notable connections between CYP2C19*2 (681 G>A) and hematological adverse effects triggered by paclitaxel (*p* = 0.032). They also identified links between CYP17 (34T>C) and both body pain and peripheral neuropathy related to paclitaxel [90]. In patients receiving docetaxel and doxorubicin, those with the ABCB1-3435 TT genotype were more prone to develop grade 3 or higher neutropenia (*p* = 0.039) and diarrhea [91]. These findings underscore the importance of pharmacogenetic factors in predicting and managing the toxicity profiles of breast cancer patients undergoing taxane-based chemotherapy.

#### 3.2.2. Cyclophosphamide (CTX)

CTX, an alkylating agent, is widely used in treating both solid and hematological malignancies, as well as for immunosuppressive therapy. As a prodrug, cyclophosphamide requires bioactivation to form active phosphoramide mustard metabolites for its therapeutic effects. The primary enzymes involved in this bioactivation are CYP2B6 and CYP2C19 [92,93].

These enzymes’ expression and activity are significantly affected by SNPs. SNPs in the coding region of CYP2B6 can alter enzyme production or activity, with these effects often being substrate-dependent. For instance, the 516G>T SNP (rs3745274) has been shown to increase enzyme activity through homotropic cooperativity in recombinant systems [94]. Yet SNPs can also induce aberrant splicing, leading to mRNA that lacks an exon and thereby reducing the production of functional protein [95,96].

A study involving 38 patients found a significant association between homozygous mutations in CYP2B6 and shorter recurrence time [97]. However, a larger retrospective cohort of 350 breast cancer patients did not find significant results concerning the CYP2B6 gene [98]. This inconsistency might be attributed to pseudogenes (CYP2B7P1), fusion alleles, and copy number variations near CYP2B6, which complicate the accurate assessment of this gene’s influence [99].

Further research involving 230 breast cancer patients identified significant associations between CYP2C19 and CYP2B6 genotypes and OS [100]. Specifically, patients homozygous for CYP2B6 516G>T and A785A>G genotypes had poorer OS (*p* = 0.04 and 0.036, respectively). Additionally, multifactor dimensionality reduction (MDR) revealed significant gene-gene interactions between CYP2C19 and CYP2B6, influencing treatment responses in breast cancer patients [101].

Moreover, a comprehensive screening study on 250 Indian breast cancer patients confirmed the association of the CYP2C19*2 allele with disease outcomes, particularly highlighting the gene-dose effect of CYP2C19*2 in relation to an increased risk of adverse reactions [102].

#### 3.2.3. Anthracyclines

Anthracyclines, including doxorubicin, epirubicin, and daunorubicin, are a class of chemotherapeutic agents widely used in breast cancer treatment [103]. These drugs exert their antitumor effects through multiple mechanisms, such as DNA intercalation, topoisomerase II inhibition, and the generation of free radicals, which collectively induce cancer cell apoptosis. In the liver, doxorubicin and other anthracyclines undergo demethylation via the CYP450 enzyme system. Another significant metabolic pathway involves aldehyde reductase, which reduces anthracyclines to their alcohol derivatives. These metabolites are typically further processed and excreted through additional metabolism or glucuronidation [104,105].

The variability in drug-metabolizing enzymes has a substantial impact on both the effectiveness and the toxicity patterns of anthracyclines. Gudur et al. found a significant association between the CYP2C19*2 (681 G>A) polymorphism and increased incidences of doxorubicin-induced nausea, vomiting, fatigue, and peripheral neuropathy [90]. Additionally, aldoketoreductase AKR1C3 IVS4-212 A>G polymorphism has been associated with lower leukocyte and neutrophil counts, along with PFS and OS [106]. A prospective cohort analysis revealed that the UGT2B7 161 C>T SNP was linked to a lower risk of grade 3–4 leukopenia, higher epirubicin elimination, and an increased risk of recurrence [107]. 

The effectiveness and side effects of drugs like anthracyclines seem to be influenced by genes associated with their transportation and elimination. For example, a study focusing on the C3435T variation of the ABCB1 gene in breast cancer patients revealed that individuals with the TT genotype experienced higher area under the curve (AUC) values and OS rates, albeit with an elevated risk of diarrhea and neutropenia [83].

The metabolism of doxorubicin and daunorubicin may be affected by single nucleotide polymorphisms in the ABCB1 gene [108,109]. However, the impact of the SNP polymorphism on tumor response following chemotherapy remains controversial [110]. Some studies have indicated that the ABCB1 TT genotype is connected with poor tumor response [111], while meta-analyses have failed to confirm a substantial link between ABCB1 variations and chemotherapy response [112]. 

Besides, the SLC22A16 T>C (rs714368) polymorphism, though not significantly affecting doxorubicin plasma concentrations, has been associated with a higher risk of neutropenia and leukopenia [113]. 

Polymorphisms in genes encoding glutathione S-transferase (GST) also contribute to anthracycline drug resistance and the incidence of related adverse events [114,115]. Deletions in GST genes, such as GSTM10 and GSTT10, have been linked to decreased rates of breast cancer recurrence and mortality [116]. The GSTP1 313A>G mutation has been identified as an independent risk factor for neutropenic hematological toxicity resulting from anthracycline/taxane chemotherapy in breast cancer patients [117,118]. These findings underscore the important role of detoxification enzymes in influencing treatment responses. 

### 3.3. Anti-HER2 Targeted Therapy

#### 3.3.1. Monoclonal Antibodies

Trastuzumab and pertuzumab, both monoclonal antibodies, are primarily utilized in the treatment of HER2-positive breast and gastric cancers [119,120,121]. The monoclonal antibody binds to the HER2 receptor and subsequently undergoes degradation through intracellular protein breakdown pathways [120]. Unlike many traditional chemotherapeutic agents, their metabolism is less influenced by CYP450 enzymes. Instead, the efficacy and metabolism of these monoclonal antibodies are significantly impacted by the HER2 gene and genes related to the immune response.

Trastuzumab and pertuzumab target the HER2 receptor protein, which is encoded by the HER2 gene. Consequently, the amplification or overexpression of the HER2 gene is directly related to the therapeutic effectiveness of these antibodies. Alterations in HER2 gene status can impact their distribution and efficacy in vivo. An analysis of an Italian population database revealed that the HER2 rs1136201-G (HER2 1173A>G) allele is associated with an increased risk of cardiac toxicity related to trastuzumab treatment [122]. Additionally, a meta-analysis identified the HER2 rs1136201 polymorphism as a potential predictor of trastuzumab-related cardiac toxicity, with a combined odds ratio (OR) of 2.43 (1.17–5.06, *p* = 0.018). Further analysis of gene dose effects indicates that having multiple variant alleles may increase the likelihood of this risk [123].

FCGR2A and FCGR3A genes encode Fc gamma receptors (Fcγ receptors), which are involved in antibody-dependent cell-mediated cytotoxicity (ADCC) reactions within the immune system [124]. Polymorphisms in these receptors can affect trastuzumab-mediated immune effects, thereby influencing its efficacy. Several studies claim that FCGR polymorphisms may serve as predictors of trastuzumab efficacy [125,126], with some reports suggesting that patients with wild-type genotypes experience more significant monocyte toxicity [127]. Studies have detected that rs1801274-G in FCGR2A (FCGR2A 519A>G) is associated with reduced efficacy [122]. In the CHER-LOB trial, which involved neoadjuvant therapy for HER2-positive breast cancer, patients with the FcγR3A 559T>G who received trastuzumab plus lapatinib demonstrated a significantly higher pathological complete response (pCR) rate compared to those receiving trastuzumab or lapatinib alone [128]. However, a well-structured clinical trial with detailed enrollment, therapy, and outcome data demonstrated a negative connection between FCGR2A and FCGR3A and the effectiveness of trastuzumab [129]. 

Moreover, patients with the ABCB1-C3435T CC genotype have an increased likelihood of developing resistance to chemotherapy/trastuzumab regimens than those with the T allele. The beneficial impact of the T-allele was confirmed and validated in a comprehensive analysis using Cox regression analysis for PFS and OS [130].

#### 3.3.2. Tyrosine Kinase Inhibitors (TKIs)

TKIs are small-molecule drugs developed for HER2-positive cancers by inhibiting HER2 receptor tyrosine kinase activity. This inhibition blocks signal transduction pathways essential for cancer cell proliferation [131]. The metabolism of different TKIs varies, primarily involving CYP2C and CYP3A enzyme families in the liver [132,133].

Lapatinib’s bioactivation is primarily dependent on overall CYP3A activity [134]. Individual variations in lapatinib metabolism are closely linked to their CYP3A activity. CYP3A4 metabolizes lapatinib to N-dealkylated lapatinib, a metabolite with heightened hepatotoxicity [135]. Therefore, higher CYP3A4 activity may increase the incidence of lapatinib-induced hepatotoxicity. Conversely, CYP3A5 and CYP3A7-mediated metabolism may reduce lapatinib-induced cytotoxicity, apoptosis, and DNA damage [135].

In vitro studies have shown that neratinib-resistant cell line variants exhibit significantly increased CYP3A4 activity. When this elevated CYP3A4 activity was inhibited using ketoconazole, a modest but significant increase in sensitivity to neratinib was observed in resistant cells. This finding suggests a potential link between CYP3A4 activity and resistance to neratinib [136].

#### 3.3.3. Antibody-Drug Conjugate (ADC)

ADCs represent a significant advancement in targeted cancer therapy, combining the specificity of monoclonal antibodies with the potency of cytotoxic agents [137]. Two notable examples in breast cancer treatment are trastuzumab emtansine (T-DM1) and trastuzumab deruxtecan (T-DXd) [138,139]. T-DM1 consists of trastuzumab linked to DM1, a microtubule inhibitor. After internalization, DM1 is released and interferes with microtubule formation, effectively halting cell division [140]. T-DXd, on the other hand, combines trastuzumab with deruxtecan, a topoisomerase I inhibitor. Upon internalization by cancer cells, deruxtecan is released intracellularly, where it inhibits topoisomerase I, disrupting DNA replication and transcription [138].

The metabolism of these ADCs involves both the degradation of the antibody component and the subsequent metabolism of the conjugated cytotoxic drugs. Following cellular endocytosis, the antibody portion undergoes lysosomal degradation, releasing the cytotoxic payload. DM1 is likely metabolized by hepatic enzymes such as CYP3A4 [141], while Deruxtecan is released through esterase-mediated hydrolysis [142].

Given these processes, ADCs may exhibit complex pharmacokinetic properties and diverse biological effects in vivo, making pharmacogenetic investigations essential; they could aid in predicting drug metabolic pathways, optimizing dosage, and mitigating the risk of adverse reactions.

## 4. Challenges and Future Directions

Despite considerable advancements in pharmacogenomics over the past two decades, the field still encounters challenges and biases. One major issue is the study of non-functional genetic variants, which are often examined in diverse patient groups without enough focus on specific disease types or patient subgroups, leading to unclear results. Additionally, there is frequently insufficient consideration of environmental and pathophysiological factors, leading to skewed results. As a result, the most successful research typically focuses on well-defined patient cohorts and investigates specific gene-drug interactions through targeted pharmacogenomic testing.

Another challenge is the unexplained variability in drug pharmacokinetics. Up to 50% of the differences in drug metabolism remain unclear at the genetic level [143,144]. Identifying sufficient gene-drug pairs to explain this variability requires large studies and substantial financial investment. Additionally, variations in patient hepatic and renal function play crucial roles in drug metabolism and excretion, adding further complexity to pharmacogenetic studies.

Conducting extensive clinical trials faces considerable obstacles, notably the challenge of controlling for confounding factors. Concomitant medication use, especially in elderly patients who often take multiple drugs, can lead to potential drug-drug interactions, affecting the accuracy of study outcomes. This limits the ability to draw clear conclusions about the efficacy and safety of pharmacogenetic-based treatments.

In clinical practice, while genetic factors (like SNPs) play a key role in plasma drug levels, other factors such as drug-drug interactions, dietary effects, and patient body composition also have a major impact. This means that genetic testing alone may not always be enough. Directly monitoring drug levels and adjusting doses accordingly through therapeutic drug monitoring (TDM) could potentially address these restrictions [145]. However, integrating pharmacokinetic analysis through blood sampling in a clinical setting introduces additional complexities, requiring precise blood sampling, quality control, and specialized laboratories, which increases costs and resource demands.

The financial burden of genetic testing is the other obstacle. While sequencing technologies have become more affordable over time, they still represent a significant cost, particularly in regions with limited healthcare budgets. Setting up the necessary laboratory infrastructure and training specialists is expensive, and in many healthcare systems, genetic tests are not covered by insurance. This puts the financial burden on patients, creating inequalities where wealthier individuals or those with good insurance have access to personalized treatments while others do not.

From the patient’s perspective, pharmacogenetic testing adds complexity to personalized care. Unlike tests for new treatments, pharmacogenetic testing doesn’t introduce entirely new treatment options but serves to optimize existing therapies by personalizing drug choice and dosage. While the idea of treatments customized to a person’s genetic profile is appealing, the extra costs may make patients question whether the testing is worth it. This makes clear communication from healthcare providers is essential. Patients should be fully involved in decisions about genetic testing, with their autonomy and values respected. Clinicians must also ensure that genetic information is used as part of the patient’s overall health picture rather than in isolation.

Addressing these challenges will require a multifaceted approach. First, reducing the cost of genetic testing through technological advancements, such as high-throughput sequencing and automation, could make testing more accessible. Public-private partnerships and government support could also help reduce costs and make genetic testing a routine part of healthcare. Improving the way patients are selected for clinical trials by including other types of data alongside genetic information could lead to better understanding and more precise treatments. For instance, combining pharmacogenomics with proteomics—the study of proteins—can enhance the understanding of how drugs interact with specific protein markers, providing a more comprehensive prediction of treatment outcomes. Similarly, integrating metabolomics, which examines the small molecules involved in metabolism, can offer further insights into individualized drug metabolism, enabling more precise dose adjustments. In oncology, pairing pharmacogenetic data with tumor-specific biomarkers, such as HER2 or BRCA mutations, allows for a dual-target approach that considers both genetic predispositions and tumor characteristics, further personalizing cancer treatments.

Moreover, integrating TDM with pharmacogenomic data should be explored to further individualize therapy, particularly when drug metabolism is highly variable or precise dosing is critical. It is also important to enhance patient education and involvement to ensure they understand the importance of following personalized treatment plans. Stronger relationships between patients and healthcare providers will foster better communication and support throughout treatment. 

Finally, future pharmacokinetic research must prioritize the development of comprehensive databases that integrate genetic, environmental, and clinical data to better understand the multifactorial nature of drug responses. Large-scale, multi-center studies with standardized methodologies, combined with advances in bioinformatics and machine learning, could aid in predicting complex gene-drug interactions and identifying novel biomarkers for drug response and toxicity.

## 5. Conclusions

While pharmacogenetic-based pharmacokinetic analysis holds great promise for personalizing drug therapy, overcoming the existing challenges requires a concerted effort to enhance the accuracy, applicability, and accessibility of pharmacogenomic testing. By addressing these issues, the field can move closer to optimizing drug efficacy and safety for every patient.

## Figures and Tables

**Figure 1 pharmaceutics-16-01407-f001:**
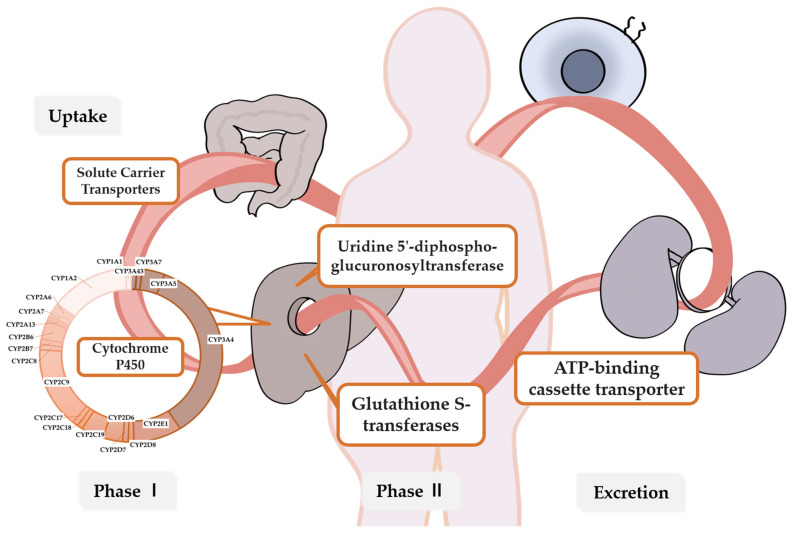
Metabolic pathways involved in drug processing. **Uptake**: Solute Carrier Transporters (SLC) facilitate the entry of drugs into liver cells for metabolism. **Phase I** Metabolism: Cytochrome P450 (CYP450) enzymes catalyze the oxidation of drugs, introducing functional groups that increase their polarity. **Phase II** Metabolism: Enzymes such as Uridine 5′-Diphospho-Glucuronosyltransferase (UGT), Glutathione S-Transferases (GST), and Sulfotransferases (SULTs) conjugate drugs with glucuronic acid, glutathione, or sulfate, enhancing their solubility and preparing them for excretion. **Excretion**: ATP-Binding Cassette (ABC) transporters expel drugs and their metabolites from liver cells into bile or blood for elimination from the body.

**Table 1 pharmaceutics-16-01407-t001:** Pharmacogenetic variants impacting pharmacokinetics in breast cancer treatment.

Drug Class	Enzyme/Genetic Variant	Effect on Pharmacokinetics
Tamoxifen (SERM)	CYP2D6	Enhanced-function allele increases symptoms and discontinuation rates; the reduced-function allele diminishes tamoxifen metabolism and efficacy.
Aromatase Inhibitors (AIs)	CYP19A1	Associated with baseline aromatase activity
	CYP2A6	Reduced-function allele elevates letrozole plasma concentrations.
	UGT1A4 and UGT2B7	Affect drug conjugation and clearance.
	GSTA1 *B*B	inhibited metabolism
CDK 4/6 Inhibitors	CYP3A4	Higher risk of toxicity with strong CYP3A4 inhibitors
Taxanes	CYP2C8*3	Higher remission rates in neoadjuvant treatment, with possible increased toxicity.
	CYP3A4	Reduced mRNA plasma level is associated with the docetaxel response rates
	ABCB1	Increased risk of neutropenia and diarrhea
	SLCO1B1 521T>C	Decreased risk of mortality
Cyclophosphamide (CTX)	CYP2B6 516G>T and A785A>G	Poorer overall survival (OS)
	CYP2C19*2	Related to an increased risk of adverse reactions (AEs)
Anthracyclines	CYP2C19*2	Increased drug-induced AEs
	UGT2B7 161 C>T	Higher epirubicin elimination, lower risk of leukopenia
	ABCB1 3435 C>T	Better OS but increased risk of diarrhea and neutropenia
	SLC22A16 T>C	Increased risk of diarrhea and neutropenia
	GSTM1 and GSTT1 deletions	Decreased recurrence and mortality rates.
	GSTP1 313A>G	Increased risk of hematological toxicity
Monoclonal Antibodies	HER2 1173A>G	Increased risk of trastuzumab-induced cardiac toxicity
	FCGR2A 519A>G	Associated with reduced trastuzumab efficacy
	FCGR3A 559T>G	Higher pCR rates with trastuzumab plus lapatinib in neoadjuvant therapy
	ABCB1 3435 C>T	Increased resistance to chemotherapy/trastuzumab regimens
Tyrosine Kinase Inhibitors (TKIs)	CYP3A4	Increased risk of lapatinib-induced hepatotoxic toxicity and associated with resistance in neratinib-resistant cells
	CYP3A5, CYP3A7	May reduce lapatinib cytotoxicity and DNA damage
Antibody-drug conjugate (ADCs)	Unknown	Needs further exploration

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
