# Peer review of "The Role of Pharmacogenetic-Based Pharmacokinetic Analysis in Precise Breast Cancer Treatment"

_pharmaceutics, 2024, doi:10.3390/pharmaceutics16111407_

Round 1

Reviewer 1 Report

Comments and Suggestions for Authors

The paper titled "The Role of Pharmacogenetic-Based Pharmacokinetic Analysis in Precise Breast Cancer Treatment" by Xinyu Wu et al addresses a critical and timely topic by exploring the integration of pharmacogenetic-based pharmacokinetic analysis into breast cancer treatment, which aligns with the growing emphasis on precision medicine. It effectively reviews the impact of genetic polymorphisms on the metabolism of commonly used breast cancer drugs, providing comprehensive insights into how these variations affect drug efficacy and toxicity. The discussion of specific genetic variants, such as those in CYP450 enzymes and transport proteins, is detailed and supported by relevant clinical data, making the findings applicable for future research and clinical application. While the review provides an extensive overview of pharmacogenetic factors, a few points need to be addressed before its publication.

  • It could benefit from a deeper exploration of the potential clinical implementation challenges, such as the cost of genetic testing and variability in access to personalized medicine.
  • The discussion assumes a high level of pre-existing knowledge from the reader, making it less accessible to a broader audience that could benefit from a more introductory overview of key pharmacogenetic concepts.
  • There is limited focus on integrating pharmacogenomics with other biomarkers or the potential for combined strategies that could further enhance treatment personalization.
  • Although the paper covers many genetic factors, the section on future directions could be expanded with specific suggestions for overcoming the identified challenges, such as strategies to reduce study costs or enhance patient stratification in clinical trials.
  • The review could provide more emphasis on patient perspectives and ethical considerations, particularly around genetic testing and the potential disparities it may introduce in treatment access.
  • Some sections, like those discussing specific drugs (e.g., tamoxifen), could benefit from additional graphical representations or summary tables that make the data more digestible and highlight key takeaways for clinicians.
Comments on the Quality of English Language

Can be improved. 

Author Response

Dear Reviewer,

Thank you for your thorough and insightful review of our manuscript. We greatly appreciate your constructive feedback, which has helped us to improve the quality and clarity of the review. We have carefully addressed each of your comments as outlined below:

Comment 1It could benefit from a deeper exploration of the potential clinical implementation challenges, such as the cost of genetic testing and variability in access to personalized medicine.

Response: Thank you for your valuable suggestion. We have expanded the discussion on the cost and access variability in genetic testing and personalized medicine. (Section 4, paragraph 5. The excerpt is as follows:

The financial burden of genetic testing is the other obstacle. While sequencing technologies have become more affordable over time, they still represent a significant cost, particularly in regions with limited healthcare budgets. Setting up the necessary laboratory infrastructure and training specialists is expensive, and in many healthcare systems, genetic tests are not covered by insurance. This puts the financial burden on patients, creating inequalities where wealthier individuals or those with good insurance have access to personalized treatments, while others do not.)

Comment 2: The discussion assumes a high level of pre-existing knowledge from the reader, making it less accessible to a broader audience that could benefit from a more introductory overview of key pharmacogenetic concepts.

Response: We have adjusted the section of discussion to make the review more accessible to a broader audience (Section 4, The main excerpt is as follows:

Despite considerable advancements in pharmacogenomics over the past two decades, the field still encounters challenges and biases. One major issue is the study of non-functional genetic variants, which are often examined in diverse patient groups without enough focus on specific disease types or patient subgroups, leading to unclear results. Additionally, there is frequently insufficient consideration of environmental and pathophysiological factors, leading to skewed results. As a result, the most successful research typically focuses on well-defined patient cohorts and investigates specific gene-drug interactions through targeted pharmacogenomic testing.

Another challenge is the unexplained variability in drug pharmacokinetics. Up to 50% of the differences in drug metabolism remain unclear at the genetic level[143, 144]. Identifying sufficient gene-drug pairs to explain this variability requires large studies and substantial financial investment. Additionally, variations in patient hepatic and renal function play crucial roles in drug metabolism and excretion, adding further complexity to pharmacogenetic studies. 

In clinical practice, while genetic factors (like SNPs) play a key role in plasma drug levels, other factors such as drug-drug interactions, dietary effects, and patient body composition also have a major impact. This means that genetic testing alone may not always be enough. Directly monitoring drug levels and adjusting doses accordingly through therapeutic drug monitoring (TDM) could potentially address these restrictions[145]. However, integrating pharmacokinetic analysis through blood sampling in clinical setting introduces additional complexities, requiring precise blood sampling, quality control, and specialized laboratories, which increases costs and resource demands.

Comment 3: There is limited focus on integrating pharmacogenomics with other biomarkers or the potential for combined strategies that could further enhance treatment personalization.

Response: A new subsection now explores the integration of pharmacogenomics with other biomarkers to enhance treatment personalization. (Section 4, paragraph 7, Line 521. The excerpt is as follows:

For instance, combining pharmacogenomics with proteomics—the study of proteins—can enhance the understanding of how drugs interact with specific protein markers, providing a more comprehensive prediction of treatment outcomes. Similarly, integrating metabolomics, which examines the small molecules involved in metabolism, can offer further insights into individualized drug metabolism, enabling more precise dose adjustments. In oncology, pairing pharmacogenetic data with tumor-specific biomarkers, such as HER2 or BRCA mutations, allows for a dual-targeted approach that considers both genetic predispositions and tumor characteristics, further personalizing cancer treatments.

Comment 4: Although the paper covers many genetic factors, the section on future directions could be expanded with specific suggestions for overcoming the identified challenges, such as strategies to reduce study costs or enhance patient stratification in clinical trials.

Response: We have revised the future directions section to include specific recommendations for overcoming clinical and research challenges. (Section 4, paragraph 7. The excerpt is as follows:

Addressing these challenges will require a multifaceted approach. First, reducing the cost of genetic testing through technological advancements, such as high-throughput sequencing and automation, could make testing more accessible. Public-private partnerships and government support could also help reduce costs and make genetic testing a routine part of healthcare. Improving the way patients are selected for clinical trials by including other types of data alongside genetic information, could lead to better understanding and more precise treatments. For instance, combining pharmacogenomics with proteomics—the study of proteins—can enhance the understanding of how drugs interact with specific protein markers, providing a more comprehensive prediction of treatment outcomes. Similarly, integrating metabolomics, which examines the small molecules involved in metabolism, can offer further insights into individualized drug metabolism, enabling more precise dose adjustments. In oncology, pairing pharmacogenetic data with tumor-specific biomarkers, such as HER2 or BRCA mutations, allows for a dual-targeted approach that considers both genetic predispositions and tumor characteristics, further personalizing cancer treatments.

Moreover, integrating TDM with pharmacogenomic data should be explored to further individualize therapy, particularly when drug metabolism is highly variable or precise dosing is critical. It is also important to enhance patient education and involvement to ensure they understand the importance of following personalized treatment plans. Stronger relationships between patients and healthcare providers will foster better communication and support throughout treatment.

Finally, future pharmacokinetic research must prioritize the development of comprehensive databases that integrate genetic, environmental, and clinical data to better understand the multifactorial nature of drug responses. Large-scale, multi-center studies with standardized methodologies, combine with advances in bioinformatics and machine learning, could aid in predicting complex gene-drug interactions and identifying novel biomarkers for drug response and toxicity.)

Comment 5: The review could provide more emphasis on patient perspectives and ethical considerations, particularly around genetic testing and the potential disparities it may introduce in treatment access.

Response: We have included a discussion on the ethical implications of genetic testing and patient engagement. (Section 4, paragraph 6. The excerpt is as follows:

From the patient’s perspective, pharmacogenetic testing adds complexity to personalized care. Unlike tests for new treatments, pharmacogenetic testing doesn't introduce entirely new treatment options but serves to optimize existing therapies by personalizing drug choice and dosage. While the idea of treatments customized to a person's genetic profile is appealing, the extra costs may make patients question whether the testing is worth it. This makes clear communication from healthcare providers is essential. Patients should be fully involved in decisions about genetic testing, with their autonomy and values respected. Clinicians must also ensure that genetic information is used as part of the patient’s overall health picture, rather than in isolation.)

Comment 6: Some sections, like those discussing specific drugs (e.g., tamoxifen), could benefit from additional graphical representations or summary tables that make the data more digestible and highlight key takeaways for clinicians.

Response: We have added summary tables to make key pharmacogenetic data more accessible to clinicians (see Section 3, Line 113, Table 1. The excerpt is as follows:

Drug Class

Enzyme/Genetic Variant

Effect on Pharmacokinetics

Tamoxifen (SERM)

CYP2D6

Enhanced-function allele increases symptoms and discontinuation rates; reduced-function allele diminishes tamoxifen metabolism and efficacy.

Aromatase Inhibitors (AIs)

CYP19A1

Associated with baseline aromatase activity

CYP2A6

Reduced-function allele elevates letrozole plasma concentrations.

UGT1A4 and UGT2B7

Affect drug conjugation and clearance.

GSTA1 *B*B

inhibited metabolism

CDK 4/6 Inhibitors

CYP3A4

Higher risk of toxicity with strong CYP3A4 inhibitors

Taxanes

CYP2C8*3

Higher remission rates in neoadjuvant treatment, with possible increased toxicity.

CYP3A4

Reduced mRNA plasma level is associated with the docetaxel response rates

ABCB1

Increased risk of neutropenia and diarrhea

SLCO1B1 521T>C

Decreased risk of mortality

Cyclophosphamide (CTX)

CYP2B6 516G>T and A785A>G

Poorer overall survival (OS)

CYP2C19*2

Related to an increased risk of adverse reactions (AEs)

Anthracyclines

CYP2C19*2

Increased drug-induced AEs

UGT2B7 161 C>T

Higher epirubicin elimination, lower risk of leukopenia

ABCB1 3435 C>T

Better OS but increased risk of diarrhea and neutropenia

SLC22A16 T>C

Increased risk of diarrhea and neutropenia

GSTM1 and GSTT1 deletions

Decreased recurrence and mortality rates.

GSTP1 313A>G

Increased risk of hematological toxicity

Monoclonal Antibodies

HER2 1173A>G

Increased risk of trastuzumab-induced cardiac toxicity

FCGR2A 519A>G

Associated with reduced trastuzumab efficacy

FCGR3A 559T>G

Higher pCR rates with trastuzumab plus lapatinib in neoadjuvant therapy

ABCB1 3435 C>T

Increased resistance to chemotherapy/trastuzumab regimens

Tyrosine Kinase Inhibitors

CYP3A4

Increased risk of lapatinib-induced hepatotoxic toxicity and associated with resistance in neratinib-resistant cells

CYP3A5, CYP3A7

May reduce lapatinib cytotoxicity and DNA damage

Antibody-drug conjugate (ADC)

Unknown

Needs further exploration

We hope that these revisions meet your expectations, and we look forward to your feedback on the revised manuscript.

Sincerely, 

Huihua Xiong

Best regards.

Reviewer 2 Report

Comments and Suggestions for Authors

Dear Authors,

Nice job!

Remark:

1)      For the phrase “precision medicine has become the cornerstone of breast cancer treatment” I would rather say “PGx could become the corner stone of breast cancer treatment” / PGx has important potential to  become become the corner stone of breast cancer treatment”  

A: NOT “precision medicine”- Because the various tools by which precision medicine seeks to achieve its goals are omics, pharmaco-omics, big data, artificial intelligence, machine learning (ML), environmental, social and behavioural factors and integration with preventive and public health. So PGx is one of the important corner stones of PM

B: NOT “… has become the cornerstone of breast cancer treatment” Because - Even though PGx is expected to determine de facto right choice of drug to the right patient at the right dose and so, to reduce outlays of therapeutic failure, its appropriate use in clinical uptake is lagging and its clinical implementation in routine medical and pharmacy practice remains challenging so far.  (particularly for Tamoxifen)

2)     For TAMOXIFEN (TX)

It would be interesting to speak selective “serotonin reuptake inhibitor (SSRI) antidepressants” (such as paroxetine) and TX interaction – Because It has been suggested that hot flashes could be resulted in tamoxifen-associated toxicities  and treated with SSRIs which are important inhibitor of CYP2D6  

[ https://www.ncbi.nlm.nih.gov/pmc/articles/PMC10065046/ ]

This interaction could be stated as

“Some studies have demonstrated important interactions between SSRs and TX causing the therapeutic effect of TX.”  etc…

Thank you!

All the best

Author Response

Dear Reviewer,

Thank you for your positive feedback and valuable suggestions. We appreciate your insights and have made the necessary revisions to address the points raised.

Comment 1: The phrase "precision medicine has become the cornerstone of breast cancer treatment" should be rephrased to reflect that PGx (pharmacogenetics) is one of the important tools of precision medicine, but not yet the cornerstone of breast cancer treatment due to its limited clinical uptake.

Response: We fully agree that PGx is one component of precision medicine, and our intention was to emphasize that precision medicine itself has become the cornerstone of breast cancer treatment, with PGx playing a critical role within that framework. To avoid any potential misunderstanding, we have revised the wording to reflect that precision medicine as a whole has transformed breast cancer care, and while PGx is crucial, its role in clinical practice is sometimes overlooked. In the following paragraph, we specifically address the underutilization of PGx in practice, as you correctly highlighted. We hope this revision better conveys the intended message. (Section 1, paragraph 2-3. The excerpt is as follows:

Precision medicine has become the cornerstone of breast cancer treatment. Routine diagnostics now integrate with molecular markers, dividing patients into Luminal, HER2-overexpressed, and basal-like[7], which guides treatment decisions on endocrine therapy, chemotherapy, or molecularly targeted therapy. By considering genetic, environmental, and lifestyle elements, precision medicine seeks to enhance the treatment effectiveness and reduce side effects.

Within this broad framework, the critical role of pharmacogenetic (PGx) variations is frequently overlooked, especially when talking about drug metabolism. Genetic factors contribute significantly to pharmacokinetic variability, particularly affecting the enzymes and transporters that regulate drug absorption, distribution, metabolism, and excretion. Understanding how an individual's genetic profile influences drug metabolism, further for its efficacy and toxicity, facilitates personalized drug selection and dosing strategies to improve effectiveness and minimize adverse reactions. )

Comment 2: It would be interesting to discuss the interaction between selective serotonin reuptake inhibitors (SSRIs), such as paroxetine, and tamoxifen, as SSRIs may inhibit CYP2D6 and impact the therapeutic efficacy of tamoxifen.

Response: Thank you for suggesting the inclusion of a discussion on the interaction between selective serotonin reuptake inhibitors (SSRIs) and tamoxifen. We have incorporated a detailed discussion of these interactions in our manuscript, referring to relevant studies, including the one you mentioned, to highlight how SSRIs may impact the therapeutic effectiveness of tamoxifen in breast cancer treatment. (Section 3.1.1, paragraph 3. The excerpt is as follows:

Furthermore, selective serotonin reuptake inhibitors (SSRIs), which are commonly prescribed to alleviate tamoxifen-induced hot flashes, are potent CYP2D6 inhibitors[37]. Studies have demonstrated that concomitant use of SSRIs, particularly paroxetine, with tamoxifen is associated with poorer clinical outcomes[37-39]. It can be assumed that CYP2D6-UM patients, who already metabolize tamoxifen at a faster rate, may experience additional inhibition from SSRIs, further impairing tamoxifen activation. )

Once again, we appreciate your constructive feedback, which has helped improve the clarity and scope of the manuscript. We hope these revisions address your concerns, and we welcome any further comments.

Sincerely,

Huihua Xiong

On behalf of all co-authors

Reviewer 3 Report

Comments and Suggestions for Authors

In principle, this review is well structured and all section are well supported by literature data.

However, there are quite a lot of typographical errors. The authors must correct the manuscript and follow the editorial style.

Author Response

Dear Reviewer,

Thank you for your kind feedback regarding the structure and content of our review. We appreciate your careful reading and your comments on typographical errors.

Comment: There are several typographical errors, and the manuscript should follow the journal's editorial style.

Response: We sincerely apologize for the typographical errors and any inconsistencies with the journal’s editorial style. We have thoroughly revised the manuscript, corrected all typographical mistakes, and ensured compliance with the journal’s formatting and style guidelines. 

We hope these revisions meet the necessary standards, and we appreciate your helpful feedback.

Sincerely,
Huihua Xiong
On behalf of all co-authors